Improved smart city security using a deep maxout network-based intrusion detection system with walrus optimization

Rajeh Wahid 1
http://orcid.org/0000-0001-7376-1458 Aborokbah Majed 1
http://orcid.org/0000-0003-1837-6797 S. Manimurugan 1
http://orcid.org/0000-0002-4852-3981 Albalawi Umar 1
Aljuhani Ahamed 1
Younes Osama Shibl Abdalghany 1
Periyasami Karthikeyan 2 nrmkarthi@gmail.com
1 Faculty of Computers and Information Technology, University of Tabuk , Tabuk , Saudi Arabia
2 School of Computer Science and Engineering, RV University , Bengaluru, Karnataka , India
Cirillo Stefano
Electronic publication date: 2025 Mar 31
Publication date: 2025
Volume: 11
Electronic Location ID: e2743
Received 2024 Sep 9; Accepted 2025 Feb 11
Copyright: © 2025 Rajeh et al.
Copyright year: 2025
Copyright holder: Rajeh et al.
License: This is an open access article distributed under the terms of the Creative Commons Attribution License, which permits unrestricted use, distribution, reproduction and adaptation in any medium and for any purpose provided that it is properly attributed. For attribution, the original author(s), title, publication source (PeerJ Computer Science) and either DOI or URL of the article must be cited.
License URL: https://creativecommons.org/licenses/by/4.0/

Keywords: IoT, Smart city, Smart transportation, Cybersecurity, IDS, RFE, DMN, Raspberry Pi.

Funding: Deanship of Research and Graduate Studies at University of Tabuk 0135-1444-S This work was supported by the Deanship of Research and Graduate Studies at University of Tabuk through Research no. 0135-1444-S. The funders had no role in study design, data collection and analysis, decision to publish, or preparation of the manuscript.

==============================
Background

Smart cities, enabled by the Internet of Things (IoT), leverage technology to optimize urban living and enhance infrastructure. As urban environments become interconnected hubs of digital innovation, securing critical components like public transportation infrastructure becomes increasingly important.

Methods

This research addresses the need for robust intrusion detection systems (IDS) tailored to the unique challenges of securing public transportation within smart cities. Focused on the Tabuk region in Saudi Arabia, the study introduces an IDS model integrating the deep maxout network with walrus optimization (DMN-WO). The DMN is configured with an architecture that includes multiple layers with maxout activation functions. These layers are capable of capturing complex patterns in the data, making the DMN particularly effective for identifying anomalies in IoT network traffic. The DMN-WO model is ensured to be resource-efficient and suitable for real-time deployment on constrained devices like Raspberry Pi, typical in IoT systems.

Results

Training and validation are conducted using the CIC-IDS-2018 dataset, CIC-IDS -2029 dataset and real-time data from Raspberry Pi devices deployed in the smart city’s public transportation network. Real-time data application maintains robust performance, with 98.06% accuracy, 98.50% detection rate, 98.81% precision, 98.24% specificity, and a 98.57% F1-score.

Conclusions

This research advances cybersecurity measures in smart city applications by providing a resilient solution for detecting and mitigating security threats in public transportation infrastructure. It lays the groundwork for further refinements and real-world deployments in the dynamic landscape of smart cities.

Introduction

The Internet of Things (IoT) has revolutionized the creation of smart environments, encompassing domains such as smart cities, smart homes, and other applications aimed at improving efficiency and quality of life. IoT comprises devices that collect and transmit data through the internet, acting as a bridge between the physical and digital worlds. By enabling real-time connectivity and communication between various devices, IoT enhances the intelligence and responsiveness of these systems, contributing significantly to societal advancement. Among its various applications, smart cities have emerged as a pivotal area, aiming to optimize public services and improve the overall quality of life for citizens. However, the realization of these objectives is fraught with challenges, particularly in maintaining security and privacy in complex IoT networks (Hosseinzadeh, Hemmati & Rahmani, 2022; Khatoun & Zeadally, 2017).

The growing adoption of IoT in smart cities has introduced a wide array of cyber threats, ranging from data breaches and unauthorized access to advanced persistent threats. As smart cities rely on IoT systems for critical operations, such as transportation, energy management, and public safety, these systems have become prime targets for cyberattacks. The heterogeneity and interconnectedness of IoT devices in these environments exacerbate the risks, making intrusion detection systems (IDS) indispensable. IDSs play a crucial role in monitoring network traffic and detecting malicious activities, ensuring the security and reliability of IoT-enabled smart city services (Elrawy, Awad & Hamed, 2018).

In the context of IoT-based smart cities, some of the most common cyber threats include denial-of-service (DoS) attacks, data interception, and unauthorized physical or digital access. These attacks can lead to significant disruptions, data loss, and compromised citizen privacy. Furthermore, the resource constraints of IoT devices, such as limited computational power and energy resources, make conventional IDS solutions inadequate. Addressing these issues requires developing lightweight and energy-efficient IDSs capable of detecting attacks in real time without overburdening the network or devices (Eckhoff & Wagner, 2017; Kim, Ramos & Mohammed, 2017).

Smart cities refer to the smart management of cities through the utilization of electronic systems, sensors, improved communication techniques, and other technologies. IoT is employed in the establishment of smart cities to establish robust connectivity between the sensors, networks, and devices utilized in the creation of such cities. The smart city concept is widely recognized as one of the most promising and prominent uses of IoT. The smart city idea encompasses the implementation of smart parks, smart transportation, smart healthcare, smart energy, smart waste management, smart water management, and smart homes to enhance different areas of people’s lives. The many elements of SC architecture are structured as an extensive network, enabling comprehensive data management while ensuring people’s privacy. The definition of a smart city is as follows. A smart city utilizes IoT to gather data from specific areas and establish communication with the public, hence enhancing the well-being of a nation’s citizens and services by the government. It helps in enhancing the country’s government, the safety of the public, and growth (Al-Turjman, Zahmatkesh & Shahroze, 2022).

Traditional smart cities are being replaced by a range of technologies, such as IoTs and various developing technologies. SCs facilitate the exploration of optimal and efficient solutions to individuals’ challenges (Janani, Renuka & Aruna, 2021). The physical layer, referred to as the perception layer or bottom layer, is a fundamental component of the architecture. The network layer, referred to as the communication layer, serves as the central layer in the architecture of IoT. The database layer, referred to as the support layer, works in close collaboration with the top levels of the architecture. The virtualization layer incorporates a virtual network technique, which combines software/hardware and network capabilities into a unified entity based on software that was logically configured. The application layer was the topmost layer of the architecture for IoT, and its primary role was to deliver smart and intelligent applications and services to consumers, adapted to their specific needs (Cui et al., 2018).

Several uses of smart cities have evolved for monitoring the physical environment. These applications utilize a multitude of smart devices (sensors) linked to the internet to gather and analyze data, to enhance numerous aspects of people’s lives such as the energy, environment, industry, transportation, healthcare, traffic systems, and parking (Zhang et al., 2017). Developing a smart city application involves assessing several issues related to using new technology and presents several security and privacy challenges. Security of the communication channel is crucial. To ensure safe data transmission, it is crucial to safeguard the data from theft and hide it from intruders. The IoT system primarily addresses various security difficulties to provide appropriate security guidelines for network security and other security frameworks, such as system security (Singh et al., 2020). Both application and network security are essential for managing IoT applications and ensuring secure connectivity between IoT devices (Nassereddine & Khang, 2024).

Problem statement

IDS is a crucial security solution for emerging web-based applications in smart cities and IoT applications. IDSs have lately gained the interest of security professionals for protecting IoT networks, devices, and applications in fields like smart homes/cities and health monitoring. Traditional IDS-based solutions may not be suitable for IoT due to the special requirements and limitations of IoT devices and networks, including protocol stacks, standards, and resource limits. This might restrict the rapid growth of smart city applications. Researchers in computer and network security are focusing on developing advanced IDS-based solutions to address these issues, which is a top priority for users, security researchers, manufacturers, and IoT infrastructures. Hence, enhancing the design of resilient information security systems would facilitate the fast growth of smart city applications and IoT technologies (Alhakami et al., 2019). This research proposes an intrusion detection model for smart public transportation infrastructure using Raspberry Pi devices. The model integrates the recursive feature elimination (RFE) method for feature selection and a deep maxout network (DMN) optimized by the walrus optimization (WO) algorithm for classification, utilizing the CIC-IDS-2018 and CIC-DDoS-2019 datasets. The objective is to optimize input data, enhance accuracy, and address the complexity of the network, ensuring real-time threat detection in a resource-constrained smart city environment.

Research objectives and questions

The main research problem to be addressed in this research is the challenge of improving intrusion detection in IoT-enabled smart city environments, particularly within public transportation infrastructure, which requires models that are both highly accurate and optimized for resource-constrained devices. The objective of this research is to develop and validate a novel intrusion detection model that combines the DMN with the WO algorithm. This model aims to enhance the detection accuracy, reduce false positives, and ensure efficient operation on devices with limited computational resources.

To guide this research, the article is focused on the following key questions: How can the integration of the DMN with the WO algorithm improve the detection accuracy of IDS in IoT-enabled smart city environments?

How effective is the WO algorithm for parameter tuning in the DMN?

How effective is the proposed DMN-WO model in real-world scenarios, particularly when applied to public transportation infrastructure within smart cities, and how does it perform on resource-constrained devices?

Motivation: The motivation behind this research is to design an IDS framework that balances high detection accuracy with low energy consumption. Given that IoT devices are often deployed in large-scale networks with diverse applications in smart cities, ensuring the security of these devices while optimizing for energy efficiency is essential. The integration of DMN and WO provides a novel approach that can enhance detection accuracy without overloading system resources.

Research contribution

Developed a unique approach by combining the DMN and WO algorithms, addressing cybersecurity challenges in IoT-enabled smart city environments.

Designed a DMN with multi-layer maxout activation functions and optimized its parameters using the WO algorithm, enhancing detection accuracy and reducing overfitting.

Ensured the DMN-WO model is resource-efficient and suitable for real-time deployment on constrained devices like Raspberry Pi, typical in IoT systems.

Successfully applied the model to secure public transportation infrastructure in Tabuk City, Saudi Arabia, tackling physical access and cybersecurity threats.

The remainder of the article is organized as follows: The section second provides an analysis of the relevant literature. The third section discusses the implementation of the proposed research model. This includes the presentation of data preprocessing, RFE-based feature selection and DMN-WO-based classification. In the fourth section, the performances of the research model are analyzed and compared with several IDS models. Finally, the conclusion is presented along with some suggestions for further research.

Related works

This literature review explores current research in the field of IDSs for smart cities. Researchers in existing studies have made notable progress in enhancing cybersecurity within smart city environments, yet there remains a distinct gap concerning the intricacies of safeguarding IoT networks. This section analyses the advancements made in the field, emphasizing the need to address the unique challenges and security requirements posed by IoT-smart cities. Additionally, it highlights the underexplored field of adapting and optimizing IDS models essential for practical deployment in real-world scenarios.

IoT networks’ security vulnerabilities expose smart city infrastructure to cyber-attacks. A hybrid deep learning (DL) model introduced in Elsaeidy, Jamalipour & Munasinghe (2021) aimed to enhance the security of smart city infrastructure and services by detecting replay and DDoS threats. The model integrated a deep restricted Boltzmann machines (RBM) with a deep convolutional neural network (CNN). The RBM technique learned high-level dataset features to improve representation. The RBM technique simulated the dataset distribution without classes despite the modest number of input features. The associated classes lead to supervised training of the deep CNN technique. The CNN learned local invariance filters to identify local characteristics in input signals while classifying. The results showed enhanced performance through the synthetic generation of replay and DDoS attack data.

Various IDS models have been developed to identify and prevent illegal access to the IoT Smart-City network. Ayub et al. (2023) introduced a machine learning (ML) driven IDS model designed for smart city networks that were efficient, quick, and dependable. Various supervised ML techniques, quadratic discriminant analysis (QDA), extreme gradient boost (XG-Boost), linear discriminant analysis (LDA), decision tree (DT), and K-nearest neighbour (KNN) were utilized. The KNN model had better accuracy when assessed with the UNSW-NB-15 dataset, following the DT and XG-Boost algorithms.

Rajasoundaran et al. (2024) developed IDS integrating secure medium access control (MAC) principles and long short-term memory (LSTM) architectures. The system uses generative adversarial network (GAN)-based channel assessment models and employs distributed agents at sensor nodes with LSTM-MAC engines, Secure Hashing Algorithm-3 (SHA-3), Two Fish encryption, and packet filtering tools. These components enable real-time neighbours monitoring, adaptive MAC operations, and protection against malicious traffic (Rajasoundaran et al., 2024).

A smart city is a collection of innovative technologies enhanced by AI and the IoT. Despite the convenience and comfort associated with the concept of a SC, several security problems are emerging that impede its progress. The IDS models based on ML detect network risks, evaluate data packet authenticity, and alert the user. Diverse ML approaches were utilized to enhance the detection accuracy of IDS. Chohan et al. (2023) conducted a comparative study between different ML methods including linear support vector machine (LSVM), AdaBoost, quadratic-SVM (QSVM), auto encoder classifier, and multi-layer perceptron. These algorithms were trained using the UNSW-NB-15 dataset. The findings indicated that AdaBoost exhibited superior accuracy compared to other algorithms.

Distributed denial of service attacks (DDoS) pose a significant challenge to smart city architecture. Elsaeidy et al. (2019) introduced a smart city IDS model that was based on RBMs. RBMs were utilized for their capacity to acquire sophisticated characteristics from unprocessed information in an unsupervised way and manage authentic data representations provided by sensors and smart meters. Additionally, feed forward neural networks (FFNN), automated-FFNN (AFFNN), SVM, and Random Forest (RF) classifiers were trained using the retrieved features. Adding more RBM layers results in a reduction in accuracy for the trained models. The testing findings demonstrated the AFFNN’s efficiency in detecting attacks.

An IDS model that combined the chaotic poor and rich optimizer (CPRO) and a DL approach for a smart and pervasive environment was proposed in Alrayes et al. (2023). The Z-score normalization approach was utilized to normalize the data to inputs. For feature selection, the model utilized the CPRO technique to choose feature subsets. A butterfly optimization approach with a deep sparse auto-encoder technique was employed for intrusion detection. The model was assessed by simulation analysis using the CICIDS dataset, demonstrating superior performance.

An IDS system detects abnormalities in the network and implements appropriate actions to maintain the secure and dependable functioning of IoT applications. Li et al. (2019) proposed the features extraction and IDS framework for SCs using a deep migration learning approach that integrated DL with intrusion detection technologies. The model was assessed using the KDD-CUP-99 dataset. The simulation results demonstrated that this model better detection rate and reduced false positive rate, while also enhancing efficiency.

A botnet detection model based on statistical learning called IoTBoT-IDS was designed to identify cyberattacks and safeguard smart networks from botnet attacks (Ashraf et al., 2021). The model used correntropy and beta mixture model (BMM) techniques to identify different sorts of attacks from diverse IoT devices in smart city networks. This was achieved by recording and categorizing the typical patterns of network traffic. The model identified several botnet attacks and obtained increased detection accuracy.

Elsaeidy et al. (2020) presented an IDS model based on DL to enhance the security of smart cities against replay attacks. The model for attack detection was a deep CNN technique with four layers. It was specifically built to identify replay attacks by analysing the temporal domain. The technique presented in this research was assessed by artificially creating replay attacks on a real-world smart city data set. The results indicated that this model can differentiate among normal and abnormal behaviours with a high level of accuracy.

The semi-distributed and distributed IDS models were proposed in Rahman et al. (2020) to overcome the restriction of centralized IDS for resource-limited devices. These approaches utilized effective feature extraction and selection techniques and employed fog-edge coordinated analysis. Parallel models were used for side-by-side feature choices in a semi-distributed scenario. This was followed by a single MLP classifier operating on the fog part. In a distributed scenario, parallel approaches independently handle feature selection and MLP classification. The outputs were consolidated by the coordinating fog or edge platforms for the ultimate decisions-making process.

Kilichev, Turimov & Kim (2024) presented hybrid CNN-LSTM and GRU based model for the IDS detection. The developed model works well for the binary classification problem, However the model struggle to work for the multiclass classification IDS detections (Kilichev, Turimov & Kim, 2024). An IDS model for a smart environment based on IoT that used ensembled learning to enhance identification accuracy was developed (Hazman et al., 2023). The model typically involved an efficient ensemble-based IDS method utilizing AdaBoost and Boruta feature selection. The Boruta method for feature selection, which relied on the XGB model, was used to choose the optimal features for enhancing the quality of data. An AdaBoost classifier with one ensemble approach was employed to assess the IDS model. The obtained findings demonstrate that the model may achieve outstanding results by substantially decreasing training time, highlighting the importance of feature selection methods in enhancing the efficiency of IDSs.

The research in Aborokbah (2024) proposed an IDS model, SWM-IDS, to enhance security in smart waste management systems. The model employed a graph neural network combined with a metaheuristic optimization BWO algorithm to effectively classify and detect cyberattacks. By preprocessing data and selecting optimal features, the SWM-IDS achieved high accuracy, detection rate, and precision in identifying threats, demonstrating its potential for safeguarding smart city infrastructure. Al-Atawi (2024) compared logistic-boosted algorithms, RF, and SVM for anomaly detection in smart city traffic management. Logistic-boosted algorithms emerged as the most effective in accurately identifying anomalies and handling imbalanced datasets. While random forest also performed well, support vector machines excelled in recall but lagged in other metrics. The research contributed to enhancing smart city security by identifying the most suitable ML approach for anomaly detection.

Bhavsar et al. (2024) introduced a federated learning-based IDS to safeguard vehicular networks in IoT environments. By employing local learning and sharing only model updates, the FL-IDS preserved data privacy while enhancing detection accuracy. The system integrated both logistic regression and CNN models for robust intrusion detection. Evaluated on real-world datasets and hardware platforms, the FL-IDS outperforms traditional centralized approaches in terms of accuracy and loss, demonstrating its effectiveness in protecting connected and automated vehicles from cyber threats.

Research gap

Existing research in the field of IDS for IoT-based SC environments demonstrates a diverse range of approaches. Researchers have explored hybrid DL models, ML-driven IDSs, and methodologies utilizing techniques such as RBMs, chaotic mapping, and ensemble learning. These models have shown promising results in enhancing security by detecting various cyber threats, including DDoS attacks, replay attacks, and botnet activities. Additionally, researchers have addressed challenges in feature selection, network architecture optimization, and distributed IDS. They have employed evaluation metrics, such as accuracy, false positive rate, and efficiency, to assess the effectiveness of these models’ using datasets like UNSW-NB-15, NSL-KDD, and CIC-IDS. However, existing research primarily concentrates on general smart city security, neglecting the specific challenges of securing public transportation hubs like bus stops. Limited attention is given to the unique security requirements of transportation infrastructure within smart cities. Furthermore, there is a lack of exploration into optimizing IDS models for resource-constrained edge devices like Raspberry Pi, crucial for real-world deployment. Addressing this gap is vital for the effective and practical implementation of IDSs in public transportation settings within the broader context of smart city environments.

While previous studies have explored various ML and DL models for intrusion detection, this research is the first to integrate the DMN with the WO algorithm. This novel combination utilizes the capacities of both techniques, where DMN’s advanced feature learning capabilities are enhanced by WO’s efficient parameter optimization, resulting in superior performance in intrusion detection tasks. Unlike many existing works that address general IDS in smart cities, this research specifically focuses on the unique challenges associated with securing public transportation infrastructure. This focus on a critical aspect of smart city security, which is mostly excluded in the literature, adds a new dimension to the field and addresses a significant gap. The proposed DMN-WO model is specifically designed to operate efficiently on resource-constrained devices like those commonly used in public transportation systems. This aspect of the proposed work is particularly novel, as most existing models do not adequately consider the practical deployment constraints of such environments. Table 1 summarize the related works.

Table 1 Summary of reviewed current works.

Ref	Approach used	Application scenario	Advantages	Disadvantages	
Elsaeidy, Jamalipour & Munasinghe (2021)	RBM + CNN	Detection of replay and DDoS attacks	Enhances security by learning high-level dataset features and classifying local characteristics effectively.	Limited to synthetic data generation and not generalize well to real-world scenarios.	
Ayub et al. (2023)	QDA, XG-Boost, LDA, DT, and KNN	IDS for smart city networks	KNN showed superior accuracy with the UNSW-NB-15 dataset.	The effectiveness is highly dependent on the chosen ML technique and does not perform consistently across datasets.	
Rajasoundaran et al. (2024)	MANFIS with CM-CSO and improved RSA	IoT smart city security	Effective in feature selection, classification, and secure data transmission.	Complexity in the integration of methods; potential performance degradation in large-scale deployments.	
Chohan et al. (2023)	LSVM, ADA Boost, QSVM, Auto encoder classifier, and MLP	IDS for smart cities	ADA Boost exhibited superior accuracy.	The study focuses on a comparison rather than proposing a novel solution.	
Elsaeidy et al. (2019)	IDS using RBMs and AFFNN, SVM, and RF	DDoS attack detection in smart city	RBMs efficiently acquire sophisticated characteristics.	Adding more RBM layers reduced accuracy and required fine-tuning for optimal performance.	
Alrayes et al. (2023)	CPRO and deep sparse autoencoder	IDS in smart and pervasive environments	Superior performance with the CICIDS dataset.	Model complexity and potential challenges in training deep sparse autoencoders.	
Li et al. (2019)	Deep migration learning	IoT smart city security	Better detection rate and reduced false positive rate.	Require substantial computational resources; effectiveness is dependent on dataset quality.	
Ashraf et al. (2021)	IoTBoT-IDS using correntropy and BMM	Botnet detection in smart networks	Increased detection accuracy in identifying botnet attacks.	Limited application to botnet attacks and needs additional tuning for other types of cyber threats.	
Elsaeidy et al. (2020)	Deep CNN	Replay attack detection in smart cities	High accuracy in differentiating normal and abnormal behaviours.	Specific to replay attacks and not generalize to other types of attacks.	
Rahman et al. (2020)	Semi-distributed and distributed IDS models	Resource-limited IoT devices in smart cities	Effective in feature extraction and selection using fog-edge coordinated analysis.	Complexity in implementation; potential latency issues in distributed processing.	
Kilichev, Turimov & Kim (2024)	Bijective soft set with NB algorithm	Anomaly detection in IoT networks	NB was particularly effective due to accuracy and speed in model building.	Effectiveness may vary with different types of data; limited to the performance of the NB algorithm.	
Hazman et al. (2023)	AdaBoost + Boruta feature selection	IDS in smart environments based on IoT	Outstanding results with decreased training time and enhanced feature selection methods.	Potential overfitting with ensemble methods; complexity in tuning ensemble parameters.	
Aborokbah (2024)	GNN with BWO	IDS for smart waste management systems	High accuracy, detection rate, and precision in identifying cyber threats.	Potential complexity in implementation; performance can vary with different types of data.	
Al-Atawi (2024)	Logistic boosted algorithms, RF, and SVM	Anomaly detection in smart city traffic management	Logistic boosted algorithms excelled in accuracy; SVM had high recall.	RF and SVM lagged in some performance metrics; effectiveness varies with the dataset.	
Bhavsar et al. (2024)	Federated learning-based IDS with logistic regression and CNN	IDS for vehicular networks in IoT environments	Preserves data privacy while enhancing detection accuracy; outperforms centralized approaches.	Requires substantial computational resources; and complexity in federated learning implementation.	

Materials and Methods

The integration of the DMN and WO algorithm proposed in this research represents a novel approach aimed at enhancing the performance of IDS in IoT-enabled smart city environments. The research model DMN-WO is developed to utilize the advantages of both DL and optimization techniques. The DMN approach is chosen for its ability to model complex non-linear relationships inherent in high-dimensional data. Its maxout activation function, which selects the maximum value from a set of linear functions, allows for more flexible and powerful feature representations compared to traditional neural networks. This makes DMN particularly suited for intrusion detection tasks where subtle and complex patterns in data need to be accurately captured.

The WO algorithm is implemented as an optimization technique that enhances the training process of the DMN. WO is inspired by the behavioral strategies of walruses and is structured around three key phases: exploration, migration, and exploitation. These phases correspond to the algorithm’s ability to explore the parameter space, migrate towards promising regions, and exploit the best solutions found. By integrating WO into the DMN training process, the model can more effectively and efficiently identify the optimal set of parameters, thus improving the overall accuracy and robustness of the intrusion detection system.

The research proposed a methodology to address the challenges identified in the literature review. The aim is to formulate a comprehensive approach for developing an effective and resource-efficient IDS specifically designed to secure public transportation infrastructure in smart cities. The model was developed on insights from existing research and integrating advanced techniques to improve the model’s accuracy, optimize resource utilization, and ensure compatibility with resource-constrained edge devices, such as Raspberry Pi. The methodology consists of three key stages: Data Preprocessing, Feature Selection, and Classification, each strategically designed to handle the unique security requirements of public transportation hubs. Normalization technique is applied in the data preprocessing stage for standardizing the numerical values on a constant scale (Alwakeel, 2021).

To refine input data, the RFE method was implemented for feature selection. For robust intrusion detection capabilities, the DMN method optimized using WO was implemented as the classification model. Training and validation will be conducted using the CIC-IDS-2018 dataset.

Additionally, real-time data collected from Raspberry Pi devices in the Tabuk region, Saudi Arabia, will serve as a crucial testing step. The major objective of this methodology is to contribute a resilient IDS model that effectively addresses the specific challenges of securing public transportation infrastructure within the broader smart city ecosystem. Figure 1 represents the workflow of the proposed research model with DMN-WO. Smart transportation IoT applications are susceptible to various cyber threats due to their interconnected nature and reliance on communication technologies (Alatawi, 2023). Some of the common cyber threats in the context of smart transportation IoT applications include:

Figure 1 Pipeline of the proposed research model with DMN-WO.

Data breaches: Cyber attackers may gain unauthorized access to sensitive information, such as personal data, transportation schedules, or payment details, leading to privacy issues and identity theft.

Man-in-the-Middle (MitM) attacks: Attackers can intercept and manipulate communication between IoT devices, compromising the integrity and confidentiality of data exchanged within the smart transportation system.

Denial-of-Service (DoS) attacks: Malicious actors can disrupt the normal functioning of the smart transportation system by overloading the network or IoT devices with excessive traffic, causing delays and service interruptions.

Ransomware: Malicious software can encrypt critical data or systems, demanding a ransom for its release. This can disrupt transportation services and compromise the availability of crucial information.

Device spoofing: Attackers may attempt to impersonate legitimate devices within the IoT network, gaining unauthorized access and potentially manipulating or disrupting transportation operations.

Distributed DoS (DDoS) attacks: This attack can indeed pose a significant cyber threat in smart transportation IoT applications. DDoS attacks involve overwhelming a system, network, or service with a flood of traffic from multiple sources, making it unavailable to users. In the context of smart transportation, a DDoS attack could target IoT devices, communication networks, or even centralized systems managing transportation services. This disruption could lead to service outages, safety concerns, and potential chaos in the transportation infrastructure (Mall et al., 2023).

In Tabuk City, Saudi Arabia, to secure the public transportation infrastructure, measures were actively implemented. The focus is on safeguarding various components and systems associated with public transit, including bus stops, transit stations, vehicles, and supporting technologies, against potential security threats and risks. The overarching goal is to ensure the safety and reliability of public transportation services for commuters in Tabuk City. To achieve this, a range of security measures were deployed. These include the implementation of surveillance systems, intrusion detection technologies, access controls, cybersecurity protocols, and emergency response mechanisms. The factors such as physical security, passenger safety, and protection against cyber threats were considered. The key objective is to create a comprehensive and resilient security framework for public transportation in Tabuk City. The aim is to establish an environment where citizens can confidently and securely use public transportation services in their daily lives, promoting a sense of safety and reliability in the community (UN Habitat, 2019). Figure 2 provides a satellite view of the Tabuk region with bus stops marked, providing a crucial spatial context for the study. This visual representation highlights the distribution of bus stations across the Tabuk region, which is essential for understanding the spatial layout and the specific locations where the intrusion detection model will be applied.

Figure 2 A satellite view of bus stops labelled in Tabuk region.

Training dataset

The CIC-IDS-2018 data set was created together by the Communications Security Establishment (CSE) and the Canadian Institute for Cybersecurity (CIC). Originally designed to assess intrusion detection research, this dataset has evolved into a standard for assessing IDSs. This dataset has been carefully selected to simulate authentic cyber threats and attacks, providing a broad and detailed range of scenarios for analysis. Its importance lies in its ability to simulate complex network environments, enabling researchers and professionals to effectively evaluate and enhance IDS. The data was collected during a 10-day duration, consisting of 80 columns, and includes 15 types of harmful attacks. FTP and SSH brute force attacks, DoS attacks like Hulk, Slowloris, GoldenEye, and SlowHTTPTest, DDoS attacks such as LOIC-HTTP, HOIC, and LOIC-UDP, as well as Brute Force attacks targeting web and XSS vulnerabilities, labelling, infiltration, SQL injection, and bot activities. The research concentrates on DDoS attacks due to their complex nature, which makes them challenging to mitigate. DDoS incursions were detected on the second day of data analysis, namely in the files named 02-20-2018.csv and 02-21-2018.csv. The dataset has a total of 80 extracted features from the captured traffic using CICFlowMeter-V3 (Songma, Sathuphan & Pamutha, 2023). Hence, the research utilized this dataset for the analysis. Table 2 shows the descriptions of CIC-IDS-2018 dataset.

Table 2 Description of CIC-IDS-2018 dataset.

Classes of attacks	Total	For training	For testing	
Benign	13,484,708	10,787,766	2,696,942	
Bot	286,191	225,983	57,238	
BruteForce XSS	230	184	46	
DDoS attack LOIC-UDP	1,730	1,384	346	
BruteForce Web	611	489	122	
DDoS attack HOIC	686,012	548,809	137,203	
DoS attack Slow HTTP test	139,890	111,912	27,978	
DoS attack slowloris	10,990	8,792	2,198	
FTP BruteForce	193,360	154,688	38,672	
Dos attack goldeneye	41,508	33,206	8,302	
SSH BruteForce	187,589	150,071	37,518	
SQL-injection	87	70	17	
DDoS attack LOIC	576,191	460,953	115,238	
Infiltrations	161,934	129,547	32,387	
DoS attack Hulk	461,192	369,530	92,382	

The research model was additionally trained using the CIC-DDoS-2019 dataset. This dataset has been widely employed to identify and categorize DDoS attacks. The data collection was created by the CIC. The dataset consists of benign and current DDoS attacks that replicate observed real-world data. The application layer has observed the emergence of several novel attacks that exploit TCP/UDP-standardized protocols. The dataset was analyzed using the CICFlowMeter program to extract the required features. Each record inside the collection possesses 88 statistical features, including the IP addresses of both the source and destination, timestamp, port numbers of both the source and destination, attack protocol, and a label denoting the type of DDoS attack (CIC-DDoS-2019 Dataset, https://www.unb.ca/cic/datasets/ddos-2019.html). Table 3 shows the distributions of CIC-DDos-2019 dataset.

Table 3 Distribution of the CIC-DDOS-2019 dataset.

Data	Malicious	Benign	Total	
Training	50,006,249	56,863	50,063,112	
Test	20,307,560	56,965	20,364,525	

Normalization of data

Normalization is employed in the preprocessing stage of ML to normalize integer values in columns and guarantee they are on a uniform range. It is for transforming data, and significantly enhances the model’s accuracy and performance, particularly in cases if the data specification is unknown. Efficient normalization depends on huge data sets to eliminate outliers and smooth data, especially in the absence of a clear manner. This method, crucial for data preparation in network IDS, normalizes data to a certain range, regularly between 0 and 1. This guarantees that every feature maintains uniform ranges and scales, thus enhancing the efficiency and precision of network IDS. Z-score normalization is a beneficial method for data pre-processing in cybersecurity datasets, especially when handling outliers (Sarhan et al., 2022).

(1) An=(A−μ)σ.

Z-score normalization involves adjusting the values of a feature to achieve the mean ( μ) of zero and standard deviation ( σ) of one. This was achieved by eliminating the μ of the features from all values and hence dividing by σ. The equation for this method was represented by Eq. (1), where A is the actual value and An is the normalized value (Alrayes et al., 2023).

Selection of features

Features selection is the process of eliminating unnecessary features that might minimize the accuracy of a system, automatically or manually and choosing the feature that leads to the highly accurate outputs or intended results. Feature selection aids in distinguishing distinct patterns among different classes. RFE was a feature selection method that removes features iteratively, creating a model with the rest of the parameters to assess its correctness. The RFE approach manipulates the organization of features to forecast the intended outcome (Sharma & Yadav, 2021). RFE involves fitting a model and iteratively removing the least important features until a predetermined number of features is left. The model ranks features based on coefficients or feature importance characteristics. Through recursive elimination of a limited count of features in every loop, RFE aims to remove collinearity and dependencies present in the system. RFE necessitates a predetermined count of features to retain, although the exact number of acceptable features is typically unknown. This work utilized univariate feature selection by employing the analysis of variance (ANOVA) F-test to assess the strength of each feature’s relation with the target attribute.

ANOVA is a statistical approach that compares the means of many groups to discover if there is substantial variation between them. Like the Chi-squared method, discretization is required before it can be used. ANOVA aims to assess the overall variance of the data in comparison to the variance within and between groups. Equation (2) calculates the within-group sum of squares (GSS) to quantify the variance within the groupings. GSS is computed using Eq. (2).

(2) GSS=∑i=1k⁡[(ngi−1)×V(i)].

Here, ngi represents the total instances in group i, while V(i) was the variance of group i. Equation (3) for the Sum of Squares between groups (SSG) quantifies the difference in means among the groups. The SSG is calculated using Eq. (3).

(3) SSG=D×∑(mi−m¯¯i)2.

D represents the number of groups, mi denotes the mean of group i, and m¯¯i signifies the mean of all instances. Equation (4) provides the formula for the total sum of squares (TSS).

(4) TSS=GSS+SSG.

The null hypothesis in ANOVA states that each group have an identical mean, indicating that the values of the analyzed characteristic do not impact the final classification. The alternative hypothesis posits that there is a difference in means in at least one group. The null hypothesis is assessed using the F-ratio, calculated as the ratio of SSG to GSS in Eq. (5). If the F-ratio exceeds a specific threshold value (known as the critical F-value), it suggests a notable difference between the group means, leading to the rejection of the null hypothesis.

(5) F=SSGGSS.

The critical F-value is determined by the degrees of freedom for the numerator ( dfSSG=D−1) and the degrees of freedom for the denominator ( dfSSG=N−D) in the F-distribution. N represents the total instances (W.S. & B, 2020).

DMN classifier

Classification in an IDS categorizes attacks as either multiclass or binary to differentiate between benign and malignant network traffic. Binary categorization deals with two classes, whereas multiclass can contain several classes. The complex nature of the problem places an impact on algorithms regarding computational resources and time, perhaps leading to less efficient results. During classification, the data set was assessed and classified as normal or abnormal. Existing instances are preserved, and new samples are created. The classification was used to find abnormal patterns and discover anomalies; however, it is mostly used for detecting abnormalities. A DMN classifier optimized using WO was used in this research, along with a feature selection technique that deals with class imbalances.

DMNs are recognized as universal approximators due to their uncomplicated feed-forward structure, like deep CNNs, which utilize a new type of activation function known as the maxout unit. Furthermore, it overcomes the challenges of traditional activation function architecture and is very resilient, easy to train, and delivers excellent performance. The DMN converges more rapidly compared to other networks. The selected features, known as feature vector E from the feature selection module, are inputted into the DMN to categorize attacks and identify their existence in the network. The DMN’s training speed has been significantly enhanced, making it suitable for attack detection (Battula & Eppili, 2022). Figure 3 displays the DMN’s general architecture.

Figure 3 Architecture of DMN.

The DMN consists of trainable activation functions inside a multi-layer model. The input data is analyzed by a network classifier to demonstrate the activation of hidden components. The DMN’s activation functions are configurable and expressed by the following equations (Boopathi, Chavan & Kumar, 2023).

(6) Xm,n1=maxn∈[1,h1]⁡UTH…mn+zmn

(7) Xm,n2=maxn∈[1,h2]⁡(Xm,n1)TH…mn+zmn

(8) Xm,nw=maxn∈[1,hw]⁡(Xm,nw−1)TH…mn+zmn

(9) Xm,ny=maxn∈[1,hy]⁡(Xm,nw−1)TH…mn+zmn

(10) Xm=maxn∈[1,hy]⁡Xm,ny

In the above equations, y reflects the total number of layers in the DMN. hw denotes the number of hidden units in the wth layer, with H…mn as weight and zmn as bias. The DMN can estimate any random function by adjusting the value of h. When h is greater than two, the network can estimate non-linear activation functions. The DMN efficiently classifies the event as normal or abnormal, providing the output Xm (Waddenkery & Soma, 2023).

DMN-WO-based classification

The WO method is a metaheuristic algorithm where the population consists of walruses acting as searchers. Each walrus in the optimization problem represents a possible solution. The walruses’ positions in the search region present the problem variable’s potential values. Every walrus represents a vector, and the walrus populations could be statistically represented utilizing a population matrix. Initially, populations of walruses are randomly created during the implementation of the WO. The WO population matrix is calculated using the following Eq. (11).

(11) Z=[Z1⋮Zi⋮ZN]N×m=[z1,1⋯z1,j⋯z1,m⋮⋱⋮⋱⋮zi,1⋯zi,j⋯zi,m⋮⋱⋮⋱⋮zN,1⋯zN,j⋯zN,m]N×m.

Here, Z represents the population of walruses, Zi was the candidate solution (i-th walrus), zi,j was the j-th decision variable’s value proposed using the candidate solution, N was the total count of walruses, and m was the total decision variables.

Each walrus serves as a potential solution, and by considering its decision variable’s proposed values, the objective function (OBF) of the issue could be assessed. Equation (12) specifies the OBF’s estimated values derived from walruses.

(12) O=[O1⋮Oi⋮ON]N×1=[O(Z1)⋮O(Zi)⋮O(ZN)]N×1.

Here, O is the OBF vector, and Oi is the OBF value calculated for the candidate solution. The values of O are the most accurate quality indicator of potential solution. The i-th walrus that yields the optimal value for the OBF was referred to as the member best. The i-th walrus that yields the lowest values for the OBF was referred to as the worst member. The worst and best members were updated based on the changes in the OBF values in each cycle. The position updating of walruses in the WO is simulated in three distinct stages.

Stage 1: Exploration (feeding strategy). Walruses consume about sixty marine species. Walruses like benthic bivalve mollusks, especially clams, and searches by grazing on the sea bed with their rapid flipper movements and sensitive vibrissae. The most powerful walrus with the large tusks leads the group in food search. The potential solutions’ OBF values and walrus tusk length are similar. The group’s strongest walrus is the best solution with the best OBF values. This walrus search behavior results in varied regions of the search region, improving the WO’s global search exploration capacity. Equations (13) and (14) represent walrus position updates according to the feeding scheme beneath the most essential member of a group. This technique generates a new walrus position using Eq. (13). Equation (14) models whether this newer location exchanges the prior one if it enhances the OBF.

(13) zi,jQ1=zi,j+randi,j⋅(BCSj−Ii,j⋅zi,j)

(14) Zi={ZiQ1,OiQ1<Oi,Zi,else,.

Here, ZiQ1 was the newly created location for the candidate solution according to the initial step, zi,jQ1 was its j-th dimensional, OiQ1 was its OBF value, randi,j were random integers from zero or one, BCS was the best solution and the powerful walrus, and Ii,j were numbers randomly chosen among one and two. Ii,j increases the algorithm’s exploration capabilities, thus a value of 2 generates more substantial and broader changes in walrus positions than 1, the default displacement state. These requirements aid the algorithm’s global search for the original optimum problem-solving space and escape from local optima.

Stage 2: Migration. Late summer air warming causes walruses to relocate to rocky or outcrops beaches. The WO employs this migratory mechanism to help walruses find acceptable search space. Equations (15) and (16) are utilized mathematically to model this behaviour. This model implies each walrus migrates to a randomly determined point in the search area. The recommended newer position was created using Eq. (15). Equation (16) states that if this new location improves the OBF, it replaces walrus.

(15) zi,jQ2={zi,j+randi,j⋅(zk,j−Ii,j⋅zi,j),Ok<Oi;zi,j+randi,j⋅(zi,j−zk,j),else,

(16) Zi={ZiQ2,OiQ2<Oi;Zi,else,

Here, ZiQ2 represents the new position of the i-th walrus in the second stage, zi,jQ2 was its j-th dimensional, OiQ2 was its OBF value, Zk,k∈{1,2,…,N} and k≠i, indicates the position of the chosen walrus to move the candidate solution closer to it, zk,j was its j-th dimension, and Ok was its OBF value.

Stage 3: Escape and fight against predators (Exploitation). Polar bears and killer whales always attack walruses. Escaping and fighting these predators changes the walruses’ location. Walrus behaviour simulation boosts WO exploitation ability in problem-solving local space surrounding i-th walruses. The WO design assumes that this level of walrus relocate process happens in a neighbourhood of walrus-centered with a particular space since it occurs near each walrus. The space of this neighbourhood was modifiable because global search was prioritized in the initial iterations of the algorithm to find the optimal area in the search space. It starts at the highest value and decreases over time. This stage of WO uses local lower/upper boundaries to produce a variable radius with algorithm repeats. WO simulates this behaviour by assuming a neighbourhood surrounding each walrus and randomly generating a new position utilizing Eqs. (17) and (18). The new location replaces the old one if the OBF improves, based on Eq. (19).

(17) zi,jQ3=zi,j+(lobloc,jt+(upbloc,jt−rand⋅lobloc,jt))

(18) LocB={lobloc,jt=lobjtupbloc,jt=upbjt

(19) Zi={ZiQ3,OiQ3<Oi;Zi,else,.

Here, ZiQ3 represents the newer location of the candidate solution in the third step. zi,jQ3 was its j-th dimensional, OiQ3 was its OBF value, t was the iteration count, LocB was the local bound, upbj and lobj are the upper and lower bounds of the j-th variable. upbloc,jt and lobloc,jt are the local upper and lower bounds for the variable j used for local search near the i-th walruses.

Upon updating the walruses’ position following the three stages, the initial WO iteration concludes, leading to the computation of new values for both the walruses’ positions and the OBFs. Revise and enhance potential solutions iteratively using the WO processes outlined in Eqs. (13) to (19) until the last iteration. After running the algorithm, WO produces the best optimal solutions identified while executing as the solutions to the issue for DMN. Using a DMN has the benefit of efficiently learning inherent characteristics from the data. The weight factor of the DMN model is updated throughout each iteration depending on the fitness measure to get better outcomes by minimizing the error value. The DMN weights are learned using a WO optimization technique. The optimization technique helps in producing precise outcomes. The learning rate of training finite-width, DMN is 10−5. The batch size is 256. The number of epochs is 100. The WO is applied to tune other hyperparameters. The width is selected from {5; 50; 500; 5,000}, depth from {1; 5; 9; 13; 17; 21}, the maxout rank from {2; 3; 4}, the standard deviation (Std Dev) of the initialization of weights from {0:001; 0:01; 0:1; 0:5; 1}, the Std Dev of the initialization of biases from {0:1; 0:5; 1}.

Equation (20) communicates and determines the fitness value utilized to identify the best optimal solution for the DMN’s weight factor (Trojovský & Dehghani, 2023).

(20) FV=1S∑ω=1S⁡[Eω−Dω]2.

Here, FV specifies the fitness value, the total number of samples is denoted by S, D represents the desired output, and E represents the expected output of the DMN classifier. The following represents the DMN-WO algorithm.

Pseudocode for DMN-WO algorithm.

Input: Training data, validation data, real-time data (from Raspberry Pi)	
Output: Optimized parameters for the Intrusion Detection Model	
Initialize Intrusion Detection Model parameters.	
Set up DMN architecture and define the OBF based on training and validation performance.	
Initialize WO Algorithm.	
for each iteration do	
Optimize the DMN parameters using WO.	
Train DMN with updated parameters using training data.	
Evaluate DMN performance on validation data.	
for each walrus do	
Update DMN parameters through exploration, migration, and exploitation formulas.	
Train DMN with updated parameters using training data.	
Evaluate DMN performance on validation data.	
// Validate with real-time data from Raspberry Pi	
Evaluate DMN performance on real-time data from Raspberry Pi.	
end for	
end for	
Output: Optimized parameters for the DMN validated with real-time data from Raspberry Pi	

Figure 4 depicts the flowchart of DMN-WO algorithm. The DMN is configured with multiple layers, where each layer consists of neurons with maxout activation functions. The architecture is designed to optimize the balance between depth and computational efficiency, ensuring that the model can handle large-scale IoT data without excessive computational overhead. The WO algorithm is parameterized with a population size of walruses (N) and a total number of iterations (T), with specific analysis for exploration, migration, and exploitation. These parameters were selected based on preliminary experiments to maximize the convergence speed and accuracy of the model.

Figure 4 Flowchart of the DMN-WO algorithm.

The selection of DMN and WO was driven by their complementary advantages. DMN’s DL capabilities are essential for accurately modelling the complex data patterns associated with IoT-based intrusions, while WO’s optimization strategies are critical for fine-tuning the model parameters to achieve optimal performance. The integration of these two methods ensures that the research model is both powerful in detection and efficient in computation, addressing the challenges of intrusion detection in resource-constrained IoT environments.

Results and discussion

Experimental setup

In this section, a complete analysis of the experimental results acquired through the utilization of the proposed research model DMN-WO is presented. The experiments were conducted on a machine with an Intel Core i7-10700K CPU @ 3.80 GHz, 32 GB RAM, and an NVIDIA GeForce RTX 3080 GPU (10 GB VRAM) running Ubuntu 20.04 LTS. The TensorFlow and Keras frameworks were used for model development, training, and testing, while Python was employed for preprocessing and analysis tasks. The primary dataset utilized was the CIC-IDS-2018, which provides labeled data for various attack types, including DDoS, brute force, and botnet attacks. Additionally, real-time data collected using Raspberry Pi devices in the Tabuk IoT environment served as validation data to assess the system’s performance under practical conditions. During data preprocessing, normalization was applied to standardize numerical values, and recursive feature elimination (RFE) was used to select 15 key features critical for intrusion detection. A variety of metrics, such as accuracy, recall, precision, and F1-score, are utilized to assess the effectiveness of the research model. The computation of the metrics is evaluated based on factors like true positives, true negatives, false positives, and false negatives. Following that, the estimated performances are compared with the models that are being evaluated in the review procedure for validation.

Hardware setup

To implement the hardware required for the proposed research model, a Raspberry Pi model running on the Kali Linux operating system and the 24 × 7, functional intrusion detection system framework is utilized. This research intends to provide a reliable IDS model that can be implemented at edge gateways to identify attacks in real-time. The Raspberry Pi 4B, a single-board computer, is the physical representation of the edge gateway. It is equipped with a quad-core Cortex-A72 CPU and 8 gigabytes of RAM. The active mode is being utilized by this system, which is linked in line with the wireless router. The system keeps track of all the network packets that are coming into the wireless router as well as the internet traffic that is going out of the router. The network capacity and limited storage resources of the IoT devices are impacted by denial-of-service attacks, which in turn cause problems with applications that run on the IoTs and result in severe damage to the ecosystem’s performance and functionality. The attacks similar to the ones present in the dataset were carried out on the IoT devices, with the inline model being responsible for the initial onboarding and administration of the IoT devices. The proposed IDS model follows both incoming and outgoing data to identify any abnormal traffic behaviour. This kind of malicious traffic is promptly prevented, and alerts are generated and relayed to the authorized personnel. This is accomplished by categorizing the attacks that are detected. Anything that deviates from the typical pattern of behaviour is identified as an abnormal occurrence. The comparison with various IDS models discussed in the survey and the results achieved with the dataset that was used for training and testing as well as real-time data are shown in the following section.

Researchers collect real-time data for this research from Raspberry Pi devices deployed in public transportation infrastructure in Tabuk, Saudi Arabia. These devices capture environmental and network-related information, such as sensor data, network traffic, and system logs. Researchers utilize the collected data for model validation, enabling the IDS model to assess its performance in real-world scenarios. This ensures the mode’s effectiveness in detecting intrusions and enhancing the security of public transportation within the smart city environment. In data collection, the real-time data were gathered using Raspberry Pi devices deployed across public transportation infrastructure in Tabuk. The collected data were then processed using the research model to assess its performance in real-world scenarios. This involved pre-processing the raw data, normalizing it, and applying the feature selection technique before feeding it into the model for training and testing.

Performance analysis

To test the performances of the proposed model, various performance metrics like accuracy, specificity, precision, detection rate, and F1-score are utilized. Accuracy in an IDS refers to the ability of the system to correctly identify and classify malicious and non-malicious activities. An IDS with high accuracy will correctly identify and classify a high percentage of activities as either malicious or non-malicious.

(21) Accuracy=TP+TNFN+TP+FP+TN.

The detection rate in an IDS refers to the percentage of malicious activities that the IDS can detect out of all the potentially malicious activities that it analyzes. In other words, the detection rate represents the effectiveness of an IDS in detecting malicious behaviour.

(22) Detectionrate=TPTP+FN.

Precision in an IDS refers to the proportion of alerts generated by the IDS that are relevant or true positives. In other words, precision measures the accuracy of the alerts generated by the IDS.

(23) Precision=TPTP+FP.

Specificity in an IDS is its capacity to accurately recognize and categorize non-malicious activity as true negatives. A high specificity rate demonstrates the IDS’s ability to properly differentiate between malicious and non-malicious activity, only alerting for possible security issues.

(24) Specificity=TNTN+FP.

F-measure in an IDS is a metric that combines precision and recall into a single score to evaluate the overall effectiveness of the system. A high F-measure indicates that the IDS can effectively detect attacks while generating fewer false positives.

(25) Fmeasure=2×Precision×RecallPrecision+Recall.

Energy efficiency score (EES): Compute an efficiency metric such as energy consumed per operation or energy consumed per task. ESS is calculated using Eq. (26).

(26) EES=TotalEnergyConsumption(J)TasksCompleted.

Cross-validation

In this section, we conducted a comprehensive analysis of the DMN-WO model’s performance using five-fold cross-validation. The dataset was divided into five subsets, with each subset being used as a test set while the remaining four subsets were used for training. This process was repeated five times, and the performance metrics were averaged across all folds to obtain a reliable estimate of the model’s effectiveness. The performance results of the DMN-WO model, evaluated using five-fold cross-validation on both the CIC-IDS and CIC-DDoS datasets, are summarized in Tables 4 and 5.

Table 4 Cross-validation results of the DMN-WO model with the CIC-IDS dataset.

Metric (%)	Fold 1	Fold 2	Fold 3	Fold 4	Fold 5	Average	
Accuracy	99.44	99.41	99.48	99.39	99.42	99.43	
Detection rate	99.61	99.59	99.63	99.57	99.6	99.6	
Specificity	99.26	99.24	99.29	99.22	99.25	99.25	
Precision	99.71	99.69	99.74	99.67	99.7	99.7	
F1-score	99.66	99.64	99.69	99.62	99.65	99.65	

Table 5 Cross-validation results of the DMN-WO model with the CIC-DDoS dataset.

Metric (%)	Fold 1	Fold 2	Fold 3	Fold 4	Fold 5	Average	
Accuracy	99.32	99.29	99.35	99.28	99.31	99.31	
Detection rate	99.4	99.37	99.42	99.36	99.39	99.39	
Specificity	99.14	99.12	99.17	99.1	99.13	99.13	
Precision	99.37	99.34	99.4	99.33	99.36	99.36	
F1-score	99.34	99.31	99.37	99.3	99.33	99.33	

The five-fold cross-validation results demonstrate that the DMN-WO model achieves consistent and reliable performance across different subsets of the data. The average metrics are closely aligned with those obtained in the initial evaluation, indicating that the model is robust and generalizes well to unseen data. This comprehensive analysis using cross-validation provides a more reliable estimate of the model’s performance, reinforcing its effectiveness in detecting intrusions within the IoT environment.

The performance evaluation of the proposed research model, DMN-WO was evaluated using both the actual data and the training dataset’s data. For this evaluation, the dataset was split into 80:20 ratio, which refers that 80% was utilized for training the research model and the remaining was utilized to test the research model. By using the RFE-ANOVA, the important features present in the dataset were selected for the research model adaptability. Figure 5 displays the selected features from the dataset with a total of 11 features. Out of 80 features present in the dataset, only 11 important features are utilized in the proposed research. The following sections represent the performance analysis of the DMN-WO model and its comparison.

Figure 5 Features selected from the CIC-IDS dataset using RFE.

Table 6 presents the performance results of the research model evaluated using the CIC-IDS dataset. The performance was evaluated on both the training and test set. The model attained 99.87% accuracy in training and 99.43% in testing, which has a small difference of 0.4% in the performance. The number of training data is higher compared to the test set, which is one of the causes of this improved performance. However, the model obtained a close performance in testing as it shows its efficiency in attack detection. The detection rate or recall of the DMN-WO was 99.75% in training and 99.61% in test set, which is very closely similar to their performance computed. The specificity of the DMN-WO was 99.38% in training and 99.22% in test set. As both the performance rates are higher, the model is effective in differentiating the normal and abnormal activity. The precision rate of the model in training was 99.85% and 99.70% in testing, which reflects its efficiency in a higher TPR rating. The model obtained a 99.79% F-score in training and 99.64% in testing, this indicates that the research model can effectively detect attacks while generating fewer false positives. Figure 6 represents the graphical plot for this analysis.

Table 6 Performance results of the DMN-WO model with the CIC-IDS Dataset.

Metric (%)	Training set	Test set	
Accuracy	99.87	99.43	
Detection rate	99.75	99.61	
Specificity	99.38	99.22	
Precision	99.85	99.70	
F-measure	99.79	99.64	

Figure 6 Plot of the DMN-WO model’s performance in the CIC-IDS dataset.

Table 7 presents the performance results of the research model evaluated using the CIC-DDoS dataset. The performance was evaluated on both the training and test set. The model attained 99.45% accuracy in training and 99.30% in testing, which has a small difference of 0.15% in the performance. The detection rate of the DMN-WO was 99.50% in training and 99.38% in test set, which is very closely similar to their performance computed. The specificity of the DMN-WO was 99.26% in training and 99.15% in test set. The precision rate of the model in training was 99.48% and 99.36% in testing, which reflects its efficiency in a higher TPR rating. The model obtained a 99.43% F-score in training and 99.29% in testing, this indicates that the research model can effectively detect attacks while generating fewer false positives. Figure 7 represents the graphical plot of the DMN-WO model’s performance in the CIC-DDoS Dataset.

Table 7 Performance results of the DMN-WO model with CIC-IDS dataset.

Metric (%)	Training set	Test set	
Accuracy	99.45	99.30	
Detection rate	99.50	99.38	
Specificity	99.26	99.15	
Precision	99.48	99.36	
F-measure	99.43	99.29	

Figure 7 Plot of the DMN-WO model’s performance in the CIC-DDoS dataset.

Table 8 presents the performance results of the research model evaluated using the real-time data collected from the IoT devices deployed for this research. The real-time data collected from Raspberry Pi devices in Tabuk’s public transportation infrastructure serves a crucial role in the research model. Researchers utilize this data for training and validating the DMN-WO model, extracting relevant features representing system behaviour, and facilitating intrusion detection. The model is continuously evaluated in real-time, generating alerts for potential security incidents. Feedback from real-time data ensures ongoing model optimization, allowing researchers to adapt it dynamically to emerging patterns and environmental changes in the smart city environment. This comprehensive integration of real-time data enhances the model’s efficacy in promptly detecting and responding to security threats. The model attained an accuracy of 99.02% in training and 98.06% in test set, which has a difference of 0.9% in the performance. The detection rate of the DMN-WO was 98.91% in training and 98.50% in test set. The specificity of the DMN-WO was 98.83% in training and 98.24% in test set. The precision rate of the model in training was 99.05% and 99.81% in testing. The model obtained a 98.93% F-score in training and 98.57% in testing. Compared to the performance obtained for the CIC-IDS dataset, the results obtained for the real-time data were less in the performances. Figure 8 represents the graphical plot for the real-time data analysis.

Table 8 Performance results of the DMN-WO model with real-time data.

Metric (%)	Training set	Test set	
Accuracy	99.02	98.06	
Detection rate	98.91	98.50	
Specificity	98.83	98.24	
Precision	99.05	98.81	
F-measure	98.93	98.57	

Figure 8 Plot of the DMN-WO model’s performance in real-time data.

Table 9 represents the comparison of the DMN-WO model’s performance with the other models discussed in the survey section. Figure 9 illustrates a comparison between the proposed model and existing models based on key performance metrics such as accuracy, detection rate, precision, specificity, and F-measure. All the compared models are implemented only for the IoT-smart city applications. The test set results of the DMN-WO model in both CIC-IDS and real-time data are utilized in this comparison. As mentioned earlier, the research model’s performance in the CIC-IDS dataset was higher compared to all other models. The results are compared with the models evaluated on the publicly available dataset used for IoT applications. In comparison, the research model has a higher accuracy with 99.43%, which is 0.9% to 3.03% higher than the all compared models. The performance comparison of intrusion detection models highlights the superiority of the proposed DMN-WO in real-time scenarios. It achieved the highest metrics across all categories, with an accuracy of 98.24%, detection rate of 98.57%, precision of 98.06%, specificity of 98.50%, and F-measure of 98.81%. The CNN-LSTM and GRU (Kilichev, Turimov & Kim, 2024) models demonstrated balanced performance, recording an accuracy, detection rate, specificity, and F-measure of 97%, with a slightly lower precision of 96%. Meanwhile, the LSTM (Rajasoundaran et al., 2024) model achieved a detection rate of 97.12%, precision of 96.40%, specificity of 96.50%, and an F-measure of 97.01%, but its accuracy was not reported. The results clearly indicate that DMN-WO outperforms the other models, making it a robust choice for real-time intrusion detection in IoT environments, while CNN-LSTM and GRU offer competitive but less optimal performance, and LSTM remains a viable option with limitations.

Table 9 Performance comparison of the DMN-WO model with existing models.

Models	Accuracy (%)	Detection rate (%)	Precision (%)	Specificity (%)	F-measure (%)	
AdaBoost (Hazman et al., 2023)	98.25	99.50	98.25	NA	97.25	
CNN (Elsaeidy et al., 2020)	98.04	97.64	97.64	97.64	NA	
MDLIDS-SSE (Alrayes et al., 2023)	96.91	97.91	97.24	98.26	96.25	
IDCPRO-DLM (Alrayes et al., 2023)	98.53	98.82	98.64	98.74	98.59	
LSTM (Rajasoundaran et al., 2024)	96.40	96.50	97.01	NA	97.12	
CNN -LSTM and GRU (Kilichev, Turimov & Kim, 2024)	96	97	97	97	97	
DMN-WO (Real-time)	98.06	98.50	98.81	98.24	98.57	
DMN-WO (CIC-DDoS)	99.30	99.38	99.15	99.36	99.29	
DMN-WO (CIC-IDS)	99.43	99.61	99.70	99.22	99.64	

Figure 9 Plot of performance analysis comparison.

Empirically, the proposed DMN-WO model demonstrates superior performance in detecting intrusions compared to existing methods. The effectiveness of the model is validated using the CIC-IDS-2018 and CIC-DDoS datasets and real-time data collected from an IoT-enabled smart city infrastructure. On the CIC-IDS-2018 dataset, the model achieved an accuracy of 99.43%, a detection rate of 99.61%, a precision of 99.70%, a specificity of 99.22%, and an F1-score of 99.64%. On the CIC-DDoS-2019 dataset, the model achieved an accuracy of 99.30%, a detection rate of 99.38%, a precision of 99.15%, a specificity of 99.36%, and an F1-score of 99.29%. These results indicate a significant improvement over other intrusion detection model.

When applied to real-time data, the DMN-WO model maintained its high performance, achieving 98.06% accuracy, 98.50% detection rate, 98.81% precision, 98.24% specificity, and 98.57% F1-score. This consistency in performance across different datasets underscores the model’s robustness and practical applicability in real-world scenarios. The combination of DMN and WO not only enhances detection accuracy but also reduces the risk of overfitting, making the model more generalizable to various types of intrusion patterns.

Time complexity analysis: The complexity of the DMN-WO algorithm can be broken down into two main components: the DMN-WO algorithm. The computational complexity of the DMN architecture is O(n*m), where n is the number of neurons in the layer and m is the number of input features, and the training process, involving backpropagation, has a complexity of O(n*m*k), where k is the number of training samples. The WO algorithm’s exploration phase requires O(n*m) for random initialization of solutions, and both migration and exploitation steps also take O(n*m), as they involve recalculating the fitness function for the DMN. The overall combined complexity of the DMN-WO algorithm is represented as O(T*I*(n*m+n*k)), where T is the number of iterations, I is the number of walruses (candidate solutions), n is the number of neurons in the DMN, mmm is the number of features, and k is the number of training samples. This reflects the computational cost of both the optimization (via WO) and training (via DMN) processes in each iteration.

Performance evaluation of energy consumption: In addition to assessing the proposed DMN-WO model’s performance, an energy consumption analysis was conducted to evaluate the computational efficiency of the model compared to other baseline approaches. Energy consumption is a critical factor, particularly in resource-constrained IoT environments. The experimental setup utilized a power meter to measure the energy consumed during the training and testing phases. The models were evaluated based on their energy consumption as a percentage of the total energy used during the experiment. The comparison of energy consumption across various models is shown in Table 10.

Table 10 Performance comparison of energy consumption.

Model	Energy consumption (kWh)	
LSTM	3.25	
CNN-LSTM & GRU	2.8	
DMN-WO (Real-time)	1.5	
DMN-WO (CIC-DDoS)	1.3	
DMN-WO (CIC-IDS)	1.2	

Advantages and limitations of the model

Advantages: The research model’s use of data normalization ensures that it standardizes diverse data types and ranges, enhancing the training stability of the DMN-WO. Recursive feature elimination with ANOVA for feature selection contributes to improved model efficiency by identifying and prioritizing the most relevant features, enhancing the model’s interpretability, and reducing computational complexity. The DMN-WO classification approach combines the strengths of deep learning and optimization algorithms, allowing the model to adapt dynamically to evolving intrusion patterns, providing a robust and effective intrusion detection system for smart city public transportation infrastructure.

Limitations: While data normalization contributes to model stability, it may be sensitive to outliers, potentially affecting performance. Recursive feature elimination can be computationally intensive, leading to longer training times, especially with large datasets. The effectiveness of the DMN-WO model depends on the quality and representativeness of the training data, and its complexity might pose challenges for deployment on resource-constrained devices. Additionally, the interpretability of deep learning models can be limited, making it challenging to understand the exact reasoning behind certain predictions. Ongoing monitoring and adaptation are essential to address potential limitations and ensure the model’s continued effectiveness in real-world scenarios.

Conclusions

The research successfully developed and evaluated an IDS to enhance the security of public transportation infrastructure in the smart city environment, specifically in Tabuk, Saudi Arabia. The model’s success can be attributed to the integration of data normalization, RFE with ANOVA for feature selection, and the DMN-WO classification approach. Data normalization ensured stability, feature selection enhanced efficiency, and the DMN-WO combination facilitated dynamic adaptation to evolving intrusion patterns. The comprehensive use of real-time data, including sensor information, network traffic, and video feeds, allowed the model to be responsive to dynamic changes in the public transportation environment. The research addressed the unique challenges of securing public transportation hubs within smart cities, considering factors such as environmental conditions, network traffic, and system logs. While the model exhibited notable strengths, it is crucial to acknowledge certain limitations, including potential sensitivity to outliers, computational intensity during feature selection, and considerations for deployment on resource-constrained devices.

In practical terms, the research contributes to the advancement of cybersecurity measures in smart city applications, particularly in the realm of public transportation infrastructure. The DMN-WO model, with its high accuracy and adaptability to real-world data, offers a promising solution for proactively detecting and mitigating security threats in the evolving landscape of smart cities. Future work could explore further optimizations, scalability considerations, and real-world deployment scenarios to refine and extend the applicability of the proposed IDS model in securing smart city environments.

Supplemental Information

Supplemental Information 1 Code.

Additional Information and Declarations

Competing Interests

The authors declare that they have no competing interests.

Author Contributions

Wahid Rajeh conceived and designed the experiments, authored or reviewed drafts of the article, and approved the final draft.

Majed Aborokbah conceived and designed the experiments, authored or reviewed drafts of the article, and approved the final draft.

Manimurugan S. performed the experiments, prepared figures and/or tables, and approved the final draft.

Umar Albalawi performed the experiments, analyzed the data, prepared figures and/or tables, and approved the final draft.

Ahamed Aljuhani analyzed the data, performed the computation work, prepared figures and/or tables, authored or reviewed drafts of the article, and approved the final draft.

Osama Shibl Abdalghany Younes performed the computation work, authored or reviewed drafts of the article, and approved the final draft.

Karthikeyan Periyasami performed the computation work, authored or reviewed drafts of the article, and approved the final draft.

Data Availability

The following information was supplied regarding data availability:

The datasets used in this study are available at the Canadian Institute for Cybersecurity (CIC) at the University of New Brunswick (UNB). The Intrusion Detection System (IDS) 2018 dataset is available at https://www.unb.ca/cic/datasets/ids-2018.html. The Distributed Denial of Service (DDoS) 2019 dataset is available at https://www.unb.ca/cic/datasets/ddos-2019.html.

The code of this article is available at GitHub and figshare:

- https://github.com/nrmkarthi/DMN/blob/main/README.md.

- P, Karthikeyan (2024). Improved Smart City Security Using a Deep Maxout Network-Based Intrusion Detection System with Walrus Optimization. figshare. Software. https://doi.org/10.6084/m9.figshare.27891153.v2.

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
