# Peer review of "Improved smart city security using a deep maxout network-based intrusion detection system with walrus optimization"

_PeerJ Computer Science, doi:10.7717/peerj-cs.2743_

## Round 0.1 · original submission · Major Revisions

I hope this email finds you well. After a thorough review of your manuscript by the assigned reviewers, I would like to inform you that, while there is potential in your work, several significant concerns have been raised regarding the experimentation and methodology.

The reviewers have pointed out that certain aspects of the experimental setup lack sufficient clarity and justification. In particular, they believe that more detailed explanations and stronger validations are necessary to support your findings. Additionally, methodological improvements have been recommended to ensure the robustness and reliability of the results.

In light of these concerns, we are requesting major revisions to the manuscript. We kindly ask that you carefully address each of the reviewers' comments in your revised submission, providing additional detail and supporting evidence where necessary.

Reviewer 1 ·

Basic reporting

The authors have submitted a revised version of the paper. I am satisfied with the changes they have made to address my comments.

Experimental design

The authors have submitted a revised version of the paper. I am satisfied with the changes they have made to address my comments.

Validity of the findings

The authors have submitted a revised version of the paper. I am satisfied with the changes they have made to address my comments.

·

Basic reporting

The paper shows a proposal for a novel DMN-based model for anomaly detection to be used in a IDS to identify DDoS attacks. The authors use a known public dataset to train and evaluate the model, showing promising results.
The literature review provided covers a number of significant publications and is sufficient.
The text is fluent and generally easy to follow. The frequent use of acronyms (not reused later in the text) force the reader to stop and recall their meaning.
The results shown are fitting with the thesis and research objectives.

Some minor notes:
- **Abstract Structure:** The abstract provides excessive detail on results, which can be overwhelming. A more concise overview, highlighting key findings and contributions, would be beneficial.
- **Performance Analysis:** The performance analysis section contains excessive detail, which can detract from the main points. A more focused discussion, highlighting key metrics and comparisons, would be preferable.

Experimental design

- **Network Security Perspective:** While the ML aspects are well-presented, the network security context is underdeveloped. The statement about novel attacks exploiting TCP/UDP is vague and lacks depth.
- **Dataset Analysis:** The authors should discuss the suitability of the chosen dataset for their research. Will the on-site Raspberry PI generate data features similar to those in the dataset? What kind of software/tool is used to collect data, what filters are used? While it is interesting to indicate the IoT platform as the target of the trained model, the authors should also consider describing how the model would be integrated in the final system.
- **Feature Selection:** The rationale for using feature selection with a deep neural network is unclear. Given the model's ability to learn patterns, feature selection might be redundant.

Validity of the findings

The results shown seem coherent with the analysis shown and are backed by the provided source code.

Additional comments

Overall, the contents of the paper seem relevant and correct.
By addressing these issues indicated in the points above, the authors can significantly improve the quality and clarity of their paper.

·

Basic reporting

1. The paper contains numerous grammatical errors and awkward phrasings that detract from readability. Examples include: "significantly enhances the model's accuracy and performance."
"Enhances" is vague without specifying "how" or "compared to what." and the word "significantly" is subjective.
We suggest specifying the exact improvement: "This normalization step improves the model's precision and reduces overfitting." etc.
The authors are strongly invited to benefit from:
- A thorough review to fix grammatical errors, improve sentence structure, and clarify vague language.
- Simplifying technical terms where possible and adding concise explanations to ensure readability.
- Removing redundancies and avoiding overly complex phrasing, aiming for direct and clear language.
2. Although the introduction explains IoT in smart cities, it fails to adequately emphasize the knowledge gap this research fills. The significance of the Deep Maxout Network and Walrus Optimization is introduced without a clear justification for their selection over other state-of-the-art methods.
3. The literature review covers relevant IDS and IoT applications, but it lacks a critical assessment of why the proposed methodology (DMN-WO) specifically advances prior work. Many referenced studies are presented superficially; for instance, there is no explanation for why certain feature selection methods were chosen over others that have shown success in the field.
4. Figures, especially Figures 2 and 3, require improvement in quality and labeling for clarity.

Experimental design

5. The methodology section lacks sufficient detail on the rationale behind using the DMN-WO model specifically. While the paper mentions the Walrus Optimization’s inspiration, it provides minimal explanation as to why this optimization is particularly suited to IDS over other algorithms.
6. Although the CIC-IDS-2018 and CIC-DDoS-2019 datasets are well-known in intrusion detection research, there is no explanation of whether and how these datasets represent the unique attributes of IoT in smart cities. Additionally, the article does not justify the use of Raspberry Pi for real-time validation.
7. The rationale for Recursive Feature Elimination (RFE) with ANOVA F-tests is not sufficiently justified. Why is this combination preferred over alternative methods, like LASSO? Further clarification on why only 11 out of 80 features were selected is necessary.

Validity of the findings

8. The results are presented with high accuracy metrics, yet there is minimal discussion on the practical implications of these findings. For instance, how do the results translate to real-world scenarios in smart cities with more complex, dynamic, and heterogeneous environments than what is presented in the datasets?
9. The authors compare DMN-WO with other models but do not rigorously discuss the trade-offs in terms of computational efficiency, scalability, or adaptability. This is critical, especially for edge devices in IoT environments with limited resources.
10. There is a lack of statistical validation for the model's performance, such as confidence intervals or significance testing for accuracy metrics across different datasets and testing conditions. This weakens the claim of robustness and generalizability.
11. Although traditional metrics (accuracy, precision, etc.) are used, the authors fail to provide any metrics or assessments that would be particularly meaningful in the context of IoT and smart city environments, such as latency, energy consumption, or resilience to network instability.

Additional comments

12. While the paper claims novelty in combining DMN with WO, this integration alone does not guarantee significant theoretical advancement without a detailed comparative analysis against recent advancements in the field.
13. The paper mentions using Raspberry Pi for real-time testing but lacks any discussion on the limitations observed during deployment, including computational resource challenges, real-time adaptability, and potential overfitting on the device.

Reviewer 4 ·

Basic reporting

1. Abstract section needed to be rewritten to understand paper novelty.
2. Introduction section is very poor. This section should contain the domain introduction, need for IDS security IoT, possible attacks and finally how your proposed work provides energy efficient attack detection in IOT
3. Kindly include problem statement, motivation and research gap at the end of the introduction section.
4. Literature survey is very poor. Represent the literature survey with latest available security papers
a. Intrusion detection using dynamic feature selection and fuzzy temporal decision tree classification for wireless sensor networks
b. An improved Harris Hawks optimizer based feature selection technique with effective two-staged classifier for network intrusion detection system
c. Machine Learning Based Intelligent RPL Attack Detection System for IoT Networks
d. Prediction of middle box-based attacks in Internet of Healthcare Things using ranking subsets and convolutional neural network
e. An intrusion detection system for securing iot based sensor networks from routing attacks
f. CAPSO: Chaos adaptive particle swarm optimization algorithm
g. Secure and optimized intrusion detection scheme using LSTM-MAC principles for underwater wireless sensor networks
5. System architecture is very poor, redraw it
6. All the equations are not integrated into main text. Provide more explanation with proper references
7. Represent your algorithms in flowchart manner
8. Provide formal analysis for your proposed algorithms
9. How many attacks instance are you considering your work? Provide details on how your proposed system identifies the attacks along with the classification accuracy
10. Results are not convincing and poorly presented. Provide more details on your experimental setup and simulation parameters employed. Kindly provide the graph taken from simulation tool
11. Authors should clearly explain how the proposed system is providing better results than existing one
12. How your system provides scalability?

Experimental design

1. Abstract section needed to be rewritten to understand paper novelty.
2. Introduction section is very poor. This section should contain the domain introduction, need for IDS security IoT, possible attacks and finally how your proposed work provides energy efficient attack detection in IOT
3. Kindly include problem statement, motivation and research gap at the end of the introduction section.
4. Literature survey is very poor. Represent the literature survey with latest available security papers
a. Intrusion detection using dynamic feature selection and fuzzy temporal decision tree classification for wireless sensor networks
b. An improved Harris Hawks optimizer based feature selection technique with effective two-staged classifier for network intrusion detection system
c. Machine Learning Based Intelligent RPL Attack Detection System for IoT Networks
d. Prediction of middle box-based attacks in Internet of Healthcare Things using ranking subsets and convolutional neural network
e. An intrusion detection system for securing iot based sensor networks from routing attacks
f. CAPSO: Chaos adaptive particle swarm optimization algorithm
g. Secure and optimized intrusion detection scheme using LSTM-MAC principles for underwater wireless sensor networks
5. System architecture is very poor, redraw it
6. All the equations are not integrated into main text. Provide more explanation with proper references
7. Represent your algorithms in flowchart manner
8. Provide formal analysis for your proposed algorithms
9. How many attacks instance are you considering your work? Provide details on how your proposed system identifies the attacks along with the classification accuracy
10. Results are not convincing and poorly presented. Provide more details on your experimental setup and simulation parameters employed. Kindly provide the graph taken from simulation tool
11. Authors should clearly explain how the proposed system is providing better results than existing one
12. How your system provides scalability?

Validity of the findings

1. Abstract section needed to be rewritten to understand paper novelty.
2. Introduction section is very poor. This section should contain the domain introduction, need for IDS security IoT, possible attacks and finally how your proposed work provides energy efficient attack detection in IOT
3. Kindly include problem statement, motivation and research gap at the end of the introduction section.
4. Literature survey is very poor. Represent the literature survey with latest available security papers
a. Intrusion detection using dynamic feature selection and fuzzy temporal decision tree classification for wireless sensor networks
b. An improved Harris Hawks optimizer based feature selection technique with effective two-staged classifier for network intrusion detection system
c. Machine Learning Based Intelligent RPL Attack Detection System for IoT Networks
d. Prediction of middle box-based attacks in Internet of Healthcare Things using ranking subsets and convolutional neural network
e. An intrusion detection system for securing iot based sensor networks from routing attacks
f. CAPSO: Chaos adaptive particle swarm optimization algorithm
g. Secure and optimized intrusion detection scheme using LSTM-MAC principles for underwater wireless sensor networks
5. System architecture is very poor, redraw it
6. All the equations are not integrated into main text. Provide more explanation with proper references
7. Represent your algorithms in flowchart manner
8. Provide formal analysis for your proposed algorithms
9. How many attacks instance are you considering your work? Provide details on how your proposed system identifies the attacks along with the classification accuracy
10. Results are not convincing and poorly presented. Provide more details on your experimental setup and simulation parameters employed. Kindly provide the graph taken from simulation tool
11. Authors should clearly explain how the proposed system is providing better results than existing one
12. How your system provides scalability?

---

## Round 0.2 · accepted · Accept

I hope this message finds you well. After carefully reviewing the revisions you have made in response to the reviewers' comments, I am pleased to inform you that your manuscript has been accepted for publication in PeerJ Computer Science.

Your efforts to address the reviewers’ suggestions have significantly improved the quality and clarity of the manuscript. The changes you implemented have successfully resolved the concerns raised, and the content now meets the high standards of the journal.

Thank you for your commitment to enhancing the paper. I look forward to seeing the final published version.

·

Basic reporting

no comment

Experimental design

no comment

Validity of the findings

no comment

Additional comments

The authors have addressed the comments well.